

# Measurement-model comparison of stabilized Criegee Intermediate and Highly Oxygenated Molecule production in the CLOUD chamber

Nina Sarnela[1], Tuija Jokinen[1], Jonathan Duplissy[1], Chao Yan[1], Tuomo Nieminen[2], Mikael Ehn[1], Siegfried Schobesberger[1,3], Martin Heinritzi[4], Sebastian Ehrhart[4], Katrianne Lehtipalo[1,5], Jasmin Tröstl[5], Mario Simon[4], Andreas Kürten[4], Markus Leiminger[6], Michael Joseph Lawler[7], Matti P. Rissanen[1], Federico Bianchi[1], Arnaud P. Praplan[8], Jani Hakala[1], Antonio Amorim[9], Marc Gonin[10], Armin Hansel[6], Jasper Kirkby[4,11], Josef Dommen[5], Joachim Curtius[4], James Smith[7], Tuukka Petäjä[1], Douglas, R. Worsnop[1,12], Markku Kulmala[1], Neil M. Donahue[13] and Mikko Sipilä[1]

[1]Department of Physics, University of Helsinki, Box 64, 00014 Helsinki, Finland.
[2]University of Eastern Finland, Department of Applied Physics, PO Box 1627, FI-70211 Kuopio, Finland.
[3]Department of Atmospheric Sciences, University of Washington, 408 ATG Bldg, Box 351640, Seattle, WA 98195, USA.
[4]Institute for Atmospheric and Environmental Sciences, Goethe University of Frankfurt, Frankfurt am Main, Germany.
[5]Laboratory of Atmospheric Chemistry, Paul Scherrer Institute, 5232 Villigen PSI, Switzerland.
[6]University of Innsbruck, Institute for Ion Physics and Applied Physics, Technikerstraße 25, 6020 Innsbruck, Austria.
[7]University of California, Irvine, Department of Chemistry, Irvine, CA 92697, USA.
[8]Finnish Meteorological Institute, P.O. Box 503, 00101 Helsinki, Finland.
[9]CENTRA, Faculdade de Ciencias da Universidade de Lisboa.
[10]Tofwerk AG, 3600 Thun, Switzerland.
[11]CERN, CH-1211 Geneva, Switzerland.
[12]Aerodyne Research, Inc., Billerica, MA 01821, USA.
[13]Carnegie Mellon University Center for Atmospheric Particle Studies, 5000 Forbes Ave, Pittsburgh Pennsylvania, 15213, USA.

*Correspondence to*: Nina Sarnela (nina.sarnela@helsinki.fi)

**Abstract.** Atmospheric oxidation is an important phenomenon, which produces large quantities of low-volatile compounds such as sulphuric acid and oxidised organic compounds. Such species may be involved in nucleation of particles and enhance their subsequent growth to reach the size of cloud condensation nuclei (CCN). In this study, we investigate α-pinene, the most abundant monoterpene globally, and its oxidation products formed through the ozonolysis in the Cosmic Leaving OUtdoors Droplets (CLOUD) chamber at CERN (the European Organization for Nuclear Research). By scavenging hydroxyl radicals (OH) with hydrogen ($H_2$), we were able to investigate the formation of Highly Oxygenated Molecules (HOM) purely driven by ozonolysis, and study the oxidation of sulphur dioxide ($SO_2$) driven by stabilized Criegee Intermediates (sCI). We measured the concentrations of HOM and sulphuric acid with a chemical ionization atmospheric pressure interface time-of-flight (CI-APi-TOF) mass spectrometer and compared the measured concentrations with simulated concentrations



calculated with a dynamic model. We found molar yields in the range of 3.5 – 6.5% for the HOM formation and 22 - 32% for the formation of stabilized Criegee Intermediates by fitting our model to the measured concentrations. The simulated time evolution of the ozonolysis products was in good agreement with measured concentrations except that in some of the experiments sulphuric acid formation was faster than simulated. The results shown here are consistent with the recently published yields for HOM formation from different laboratory experiments. Together with the sCI yields, these results help to understand atmospheric oxidation processes better and make the reaction parameters more comprehensive for broader use.

## 1. Introduction

Atmospheric new-particle formation begins when trace gases form small molecular clusters, which can grow to larger sizes through the condensation of vapours. When they have reached a large enough diameter, these particles can act as a cloud and ice condensation nuclei that may affect the optical properties of clouds or have other effects on climate and air quality such as decrease in visibility. A lot of effort have been put into identifying the vapours responsible for nucleation and growth of the particles. Various studies have identified sulphuric acid and low volatility organic compounds as the key compounds in atmospheric new particle formation (Bianchi et al., 2016; Kirkby et al., 2016; Riccobono et al., 2014; Riipinen et al., 2011; Sihto et al., 2006; Tröstl et al., 2016; Weber et al., 1997; Wehner et al., 2005). Other important aerosol precursors identified in several laboratory studies include ammonia and amines (Almeida et al., 2013; Ball et al., 1999; Murphy et al., 2007). Laboratory measurements (Almeida et al., 2013; Berndt et al., 2010, 2014b; Jen et al., 2014; Kirkby et al., 2011) and computational studies (Kurtén et al., 2008; Paasonen et al., 2012) suggest that ammonia and amines can enhance particle formation but it is challenging to estimate their importance in the ambient atmosphere without comprehensive measurements of their concentration in the atmosphere. Neutral sulphuric acid – amine clusters have been observed in the CLOUD chamber experiments (Kürten et al., 2014) but similar neutral clusters have not yet been detected in the atmosphere. Field studies suggest that iodine oxides could be the key compounds for new particle formation in coastal areas during periods when high tidal movements expose algae beds to sunlight (O'Dowd et al., 2002; Sipilä et al., 2016). However, these iodine oxides do not appear as abundantly in the atmosphere as sulphuric acid or low volatile organic vapours, so their importance seem to be limited to coastal areas.

Sulphuric acid is linked with new particle formation events all around the world (Kulmala et al., 2004). Gas phase sulphuric acid was previously thought to be formed solely via OH-radical oxidation of sulphur dioxide ($SO_2$) and dimethylsulfide (Lucas and Prinn, 2005). However, stabilized Criegee Intermediates (sCI), formed in a reaction between unsaturated hydrocarbons and ozone, are also capable of oxidising $SO_2$ into sulphuric acid. While sCI's (Chuong et al., 2004; Donahue et al., 2011; Drozd and Donahue, 2011) and their reactions with $SO_2$ (Cox and Penkett, 1971) have been investigated for decades, the atmospheric relevance of sCI+$SO_2$ was demonstrated more recently (Mauldin III et al., 2012; Welz et al., 2012). In the reaction between ozone and alkenes, a primary ozonide is formed which decomposes quickly to a carbonyl and a carbonyl oxide known as the Criegee Intermediate (Criegee, 1975). A fraction of the Criegee Intermediates can be collision stabilized to form sCI (Donahue et al., 2011). In the case of α-pinene and other endocyclic alkenes, both functional groups—the carbonyl and Criegee Intermediate—remain in the same molecule. Recent studies indicate that the sCI can have a significant role in ambient sulphuric acid formation (Boy et al., 2013; Welz et al., 2012; Yao et al., 2014). Determining the reaction





rate constants for sCI + SO$_2$ reactions has been challenging and the previous estimates have varied considerably due to the lack of direct measurements of sCI compounds (Johnson and Marston, 2008). Recent studies (Berndt et al., 2012; Mauldin III et al., 2012; Welz et al., 2012) with new experimental methods have shown up to four orders of magnitude higher reaction rate constants for the reaction between a sCI and SO$_2$ compared to previous

estimates. Also differences between the reactivity of sCI derived from different alkenes and their reactivity towards SO$_2$, water and several other atmospheric compounds has been emphasized (Berndt et al., 2014a; Sipilä et al., 2014; Taatjes et al., 2013; Vereecken et al., 2012, 2014).

The other important reaction pathway associated with ozonolysis of alkenes, such as α-pinene, is the formation of Highly Oxygenated Molecules (HOM, Ehn et al., 2014; Kirkby et al., 2016; Tröstl et al., 2016). Crounse et al.

(2013) suggested that autoxidation, in which the compound is oxidised by atmospheric oxygen, plays an important role in the atmospheric oxidation of organic compounds. Organic radicals, including radicals formed when Criegee Intermediates decompose, will react with molecular oxygen (O$_2$) to form an peroxy radical (RO$_2$). The RO$_2$ can subsequently undergo an H-shift, which will be followed by subsequent O$_2$ addition to form a more oxidized RO$_2$. According to the mechanism introduced by Ehn et al. (2014), the RO$_2$ can undergo several additional reactions

with O$_2$, which eventually leads to the formation of HOM, also referred to as Extremely Low Volatility Organic Compounds (ELVOC, Donahue et al., 2012; Ehn et al., 2014; Jokinen et al., 2015) or Highly Oxidixed Multifunctional organic compound (HOM, Ehn et al., 2012). Here we call them HOM, as it was recently recognized, that not all HOM nesessarily are extremely low volatile (Tröstl et al., 2016). The RO$_2$ can also react with nitrogen oxide (NO), hydroperoxyl radical (HO$_2$) or another RO$_2$, which will terminate the autoxidation

reaction chain and form a closed shell product. The molar yield of HOM formed from α-pinene and ozone is reported to be around 3-7% (Ehn et al., 2014; Jokinen et al., 2015; Kirkby et al., 2016).

In Earth's atmosphere α-pinene is the most abundant monoterpene having yearly emissions of 50 Tg globally (Guenther et al., 1995; Seinfeld and Pankow, 2003) and around 80% of the emitted α-pinene undergo oxidation via ozonolysis (Griffin et al., 1999). The high yields of HOM acting as condensing vapours can explain a large

portion of the formed secondary organic aerosol (SOA) at least in the forested regions. At atmospheric pressure, ozonlysis of the endocyclic α-pinene generates sCI with a low but important yield, measured to be around 15% (15% (Drozd and Donahue, 2011), 15% ± 7% (Sipilä et al., 2014)).

In this study we conducted pure ozonolysis experiments in which OH was removed by a scavenger in the CLOUD chamber facility at CERN (Kirkby et al., 2011; Duplissy et al., 2016) during the CLOUD7 campaign in fall 2012.

We examined the formation of sulphuric acid originating from Criegee Intermediate oxidation and of HOM from α-pinene oxidation and compared the temporal trends of the measured to the modelled concentrations. The modelling of HOM concentration was based on the experimental yield terms obtained from recent studies by Ehn et al. (2014) and Jokinen et al. (2015) while the sulphuric acid concentration was modelled using the reaction coefficient and the yield term from the study by Sipilä et al. (2014). In addition, we calculated the yield terms for

sCI and HOM formation in the CLOUD experiments by fitting our model to the measured sulphuric acid and HOM concentrations.



## 2. Methods

### 2.1 Experiments

We conducted the experiments in the CLOUD chamber, which is a 26.1 m$^3$ electro-polished stainless steel cylinder at CERN (Geneva, Switzerland, Kirkby et al., 2011; Duplissy et al., 2016). We compared our results to previous
experiments of α-pinene ozonolysis conducted in the Tropos Laminar Flow Tube (Berndt et al., 2005) and the Jülich Plant Atmosphere Chamber facility (Mentel et al., 2009)(Table 1). In contrast to these experimental facilities, the CLOUD chamber has a smaller wall loss rate (e.g. around 1.8×10$^{-3}$ s$^{-1}$ for sulphuric acid), which is similar to the condensation sink in relatively unpolluted ambient environments. This feature allows us to investigate nucleation and growth processes with precursors at atmospherically relevant concentrations.

Table 1. Description of different experimental systems compared in this study

|  | TROPOS-LFT | JPAC | CLOUD |
|---|---|---|---|
|  | Tropos Laminar Flow Tube, Leipzig, Germany (Berndt et al., 2005) | Jülich Plant Atmosphere Chamber, Jülich, Germany (Mentel et al., 2009) | The Cosmics Leaving Outdoor Droplets, Geneva, Switzerland (Kirkby et al., 2011) |
| Description | Laminar flow glass tube with 40s residence time | Borosilicate glass chamber with 45min residence time | Stainless steel chamber with 3h resindence time |
| Volume | 0.025 m$^3$ | 1.45 m$^3$ | 26.1 m$^3$ |
| Temperature | 293 K | 289 K | 278 K |
| RH | 25-50% | 63% | 38% |
| Wall loss | 10-27% | 11×10$^{-3}$ s$^{-1}$ | 1.2-1.9×10$^{-3}$ s$^{-1}$ |
| Scavenger ( for OH) | H$_2$, propane | CO | H$_2$ |

For this study, only α-pinene ozonolysis experiments fulfilling certain conditions were selected:
• Only ozone, α-pinene and sulphur dioxide were added as precursors to the chamber
   • H$_2$ was used as OH scavenger
   • Ions were constantly removed from the chamber (i.e. neutral conditions)
   • 38% relative humidity and 278 K temperature

We used two electrodes operating at voltages of +/-30 kV inside the chamber to produce an electric field of 20
kV/m throughout the chamber which removed all the ions in order to maintain neutral conditions. All the experiments were done at 278 K and the thermal insulation kept the temperature stable within 0.05 K. The relative humidity was kept at 38% during all the experiments. The synthetic air used in the chamber was provided from cryogenic liquid N$_2$ and O$_2$ (79:21, volume ratio) and 0.1% of H$_2$ was added to the air to scavenge all the hydroxyl radicals (OH) and prevent any OH-initiated reactions. The ozone mixing ratio was kept around 22 ppbv in all the





experiments. Sulphur dioxide was added to the chamber at a mixing ratio around 70 ppbv in four experiments and at a mixing ratio of 17 ppbv in one experiment. α-Pinene was supplied with mixing ratios varying between 80 pptv and 600 pptv from a temperature controlled evaporator using $N_2$ as a carrier gas. Two counter-rotating stainless steel fans are mounted inside the chamber to achieve efficient turbulent mixing of the gases and ions (Voigtländer
et al., 2012). The total flow through the chamber is kept constant during the experiments.

We started the ozonolysis experiments with a constant concentration of $SO_2$, $O_3$ and $H_2$ in the chamber (background measurement). Then we injected α-pinene into the chamber with a constant flow rate during the whole experiment (4-7 hours). In between the experiments, the chamber was cleaned by cutting the α-pinene flow and flushing the chamber with pure air (mixture of evaporated liquid nitrogen and liquid oxygen). All formed particles were
removed by repeated charging the particles and applying the high-voltage clearing electric field inside the chamber. The conditions of each experiment are shown in Table 2.

**Table 2: The measured concentrations of precursor vapours (ozone, α-pinene and sulphur dioxide), formation rates at 2.5 nm, growth rates of sub-3nm particles and calculated yields for sCI and HOM during the experiments.**

|   | $O_3$ (ppbv) | α-pinene (pptv) | $SO_2$ (ppbv) | Formation rate ($cm^{-3}$ $s^{-1}$) | Growth rate (nm $h^{-1}$) | sCI yield (%) | HOM yield (%) |
|---|---|---|---|---|---|---|---|
| 1 | 22 | 80 | 72 | 13.26 | 1.88 | 22 | 5 |
| 2 | 24 | 80 | 72 | $9.11 \times 10^{-2}$ | 1.50 | 22 | 3.5 |
| 3 | 22 | 600 | 67 | 47.98 | 7.21 | 23 | 6 |
| 4 | 22 | 170 | 68 | 3.95 | 2.32 | 24 | 5.5 |
| 5 | 22 | 530 | 17 | 18.10 | 3.77 | 32 | 6.5 |

### 2.2 Instruments

A Proton Transfer Reaction Mass Spectrometer (PTR-MS, Ionicon Analytik GmbH, Lindinger et al., 1998) was used to measure the concentrations of volatile organic compounds (including α-pinene). The neutral particle size distribution of 2-40 nm particles was measured with a Neutral cluster and Air Ion Spectrometer (NAIS, Mirme
and Mirme, 2013). The particle size distribution of 5 to 80 nm particles was measured with a nano Scanning Mobility Particle Sizer (nanoSMPS, Wang and Flagan, 1990) and the condensation sink due to particles in the chamber was calculated from the size distribution. Sulphur dioxide concentration was measured with a high sensitivity pulse fluorescence analyzer (model 43i-TLE; Thermo Fisher Scientific Inc), and ozone with a UV photometric ozone analyzer (model 49C, Thermo Environmental Instruments).

The gas-phase sulphuric acid and HOM were detected with a nitrate ion-based Chemical Ionization Atmospheric Pressure interface Time of Flight mass spectrometer (nitrate-CI-APi-TOF, Tofwerk AG, Thun, Switzerland and Aerodyne Research Inc., USA, Jokinen et al., 2012; Junninen et al., 2010). A soft X-ray source (Hamamatsu L9490) was deployed to ionize nitric acid to nitrate ions $((HNO_3)_{0-2}NO_3^-)$, which were used as the reagent ions for



the chemical ionization. The ionization method is selectively suited for detecting strong acids such as sulphuric acid or methane sulfonic acid (Eisele and Tanner, 1993). In the case of the oxidised organic compounds, it requires molecules to have at least two hydroperoxy (OOH) groups or some other H-bond donating groups to be ionized (Hyttinen et al., 2015). A previous study of cyclohexene ozonolysis showed that in contrast to highly oxygenated products such as $C_6H_8O_7$ and $C_6H_8O_9$ (with three carbonyl groups and two and three hydroperoxy groups, respectively) products like $C_6H_8O_5$ (three carbonyl groups and one hydroperoxy group), could not be detected (Rissanen et al., 2014). However, in previous α-pinene experiments oxidised products with a O:C ratio of as low as 0.6 have been detected (Jokinen et al., 2015; Praplan et al., 2015).

The concentration of sulphuric acid was calculated according to equation 1, where a calibration coefficient $c$ is applied on the count rates of the bisulphate ion and its cluster with nitric acid normalized to the sum of count rates of reagent ions (Jokinen et al., 2012). To obtain the calibration coefficient $c$, the instrument was calibrated for sulphuric acid with a calibration setup described by Kürten et al. (2012). The calibration constant was measured to be $5\times10^9$ molecules/cm³. Taking sample tube losses into account a value of $1.25\times10^{10}$ molecules/cm³ was obtained for $c$.

$$\left[H_2SO_4\right] = \frac{HSO_4^- + (HNO_3)HSO_4^-}{NO_3^- + (HNO_3)NO_3^- + (HNO_3)_2 NO_3^-} \times c \tag{1}$$

In the experiments the concentration of sulphuric acid clusters was low since there were no stabilizing agents such as amines or ammonia added into the chamber. Thus the vast majority of the sulphuric acid concentration was in form of a monomer, not in the clusters as "hidden sulphuric acid" (Rondo et al., 2016). At most, less than 2% of the total sulphuric acid concentration was involved in the clusters while most of the time no sulphuric acid clusters were detected.

### 2.3 Estimation of HOM sensitivity

In this study, we counted all the α-pinene oxidation products that were detected and identified with nitrate-CI-APi-TOF as HOM. The total concentration of HOM was calculated by summing up the high resolution fitted signals of identified highly oxygenated compounds (see the full list of peaks in the Appendix). These compounds were detected in the range of 220 – 620 Th and their O:C ratios were between 0.6 and 1.3. Most of the elemental compositions found in the experiments were the same as have been published by Ehn et al., (2012) and Jokinen et al. (2014). The sum of signals was divided by reagent ion signals and multiplied by the same calibration constant that was used for sulphuric acid (Eq. (2)).

$$\left[HOM\right] = \frac{\sum HOM \cdot NO_3^-}{NO_3^- + (HNO_3)NO_3^- + (HNO_3)_2 NO_3^-} \times c \tag{2}$$

Since we did not have a direct calibration method for HOM, we considered three additional terms, which may affect the detection of molecules before the calibration constant of sulphuric acid can be used (Eq. (2), (Kürten et al., 2014)).





$$[\text{HOM}] = \frac{k_{SA}}{k_{HOM}} \times \frac{T_{SA}}{T_{HOM}} \times \frac{e_{SA}}{e_{HOM}} \times \frac{\sum \text{HOM} \cdot \text{NO}_3^-}{\text{NO}_3^- + (\text{HNO}_3)\text{NO}_3^- + (\text{HNO}_3)_2 \text{NO}_3^-} \times c \qquad (3)$$

The first term $k_{SA}/k_{HOM}$ corrects for the difference in reaction rate between the HOM and the reagent ions compared to sulphuric acid and the reagent ions. In the chemical ionisation method, there is an excess of nitric acid in the drift tube, where the sample flow and reagent ions meet. The nitrate dimer, $\text{HNO}_3\text{NO}_3^-$, is an extremely stable

cluster, which means that if there are some other clusters forming with $\text{NO}_3^-$ in the drift tube, they need to be even more stable than the nitrate dimer. As we can detect a large total signal of HOM-nitrate clusters, we can assume that they are very stable. If we assume that all the HOM that collide with nitrate ions in the drift tube form clusters and stick together subsequently, we get the lower limit of HOM concentration from our measurements. If all collisions would not in reality produce clusters or if some fraction of the clusters would decompose in the drift

tube or inside the high vacuum region of the TOF, the real concentrations of HOM would be higher than assumed by this method. Ehn et al. (2014) reported calculated collision limited reaction rates of $k_{HOM} = (1.5\text{--}2.8) \times 10^{-9} \text{ cm}^3 \text{ s}^{-1}$ for HOM and $k_{SA} = (1.5\text{--}2.5) \times 10^{-9} \text{ cm}^3 \text{ s}^{-1}$ for sulphuric acid. The collision limited reaction rates are so close to each other that we approximated the term $k_{SA}/k_{HOM}$ to be 1.

The second term $T_{SA}/T_{HOM}$ describes the differences in the transmission efficiency of different sized molecules or

clusters through the sampling line, as increasing size of the molecule or cluster implies smaller diffusivity. A third term $e_{SA}/e_{HOM}$ takes into account the mass discrimination effects inside the mass spectrometer. The total effect of the terms $T_{SA}/T_{HOM}$ and $e_{SA}/e_{HOM}$ was determined experimentally with a high resolution differential mobility analyser (HR-DMA) method (Junninen et al., 2010). By this method, trioctylmethylammonium bis(trifluoromethylsulfonyl)imide particles were produced with an electrospray, and size ranges were selected with

a high-resolution Vienna type Differential Mobility Analyzer (UDMA, Steiner et al., 2010) and the selected size range was guided to the APi-TOF. To calculate the transmission the signal in mass spectrometer was divided by the signal in electrometer. The transmission in the mass range between 90 and 600 Th varied so that the largest difference compared to the transmission of sulphuric acid was 1.4-fold at 320 Th ($7.3 \times 10^{-4}$). Since HOM could be measured over a wide mass range, the transmission varied between individual HOM molecules (($6.4\text{-}10.4) \times 10^{-4}$).

The averaged difference of the transmissions was around 30% so that the transmission of HOM signals was higher than the sulphuric acid signals, and this was taken into account in the concentration calculations by correcting the values according the transmission curve.

We estimated a systematic uncertainty of +50%/-33% for the sulphuric acid concentration (Kirkby et al., 2016). The estimation is based on the uncertainty of the sulphuric acid calibration and a comparison with the sulphuric

acid concentration measured by an other CIMS instrument (independently measured sulphuric acid concentration at CLOUD experiments (Kürten et al., 2011)). For the HOM concentration the uncertainty is larger due to lack of a direct calibration method. We estimated an uncertainty of +100%/-50% for HOM concentrations taking the sulphuric acid calibration, charging efficiency, mass dependent transmission efficiency calibration and sampling line losses into consideration (Jokinen et al., 2015; Kirkby et al., 2016).



### 2.3 The simulations of sCI and HOM concentrations

The temporal behaviour of the reaction products from monoterpene ozonolysis in the CLOUD chamber was simulated with a 0-dimensional dynamic model. The production of stabilized Criegee Intermediates was calculated from the measured α-pinene and ozone concentrations using a reaction rate coefficient of $8.05\times10^{-17}$ cm$^3$s$^{-1}$

(Atkinson et al., 2006). Since the temperature of CLOUD experiments was lower than in previous experiments done in TROPOS-LFT and JPAC (Table 1) we used a lower reaction rate coefficient ($8.66\times10^{-17}$ cm$^3$s$^{-1}$ in JPAC experiments and $1.1\times10^{-16}$ cm$^3$s$^{-1}$ in TROPOS-LFT experiments). The reaction of sCI with SO$_2$ is competed with the reaction of sCI with water vapour, thus, three loss paths were taken into account for the sCI: (1) its reaction with sulphur dioxide ($k_{sCI+SO2}$), (2) the thermal decomposition of sCI ($k_{dec}$) and (3) its reaction with water vapour

($k_{sCI+H2O}$). The latter two reactions are included in the loss term $k_{loss}$ (Eq. (4)). The condensation sink, wall loss and dilution are negligible compared to the loss term $k_{loss}$. The reaction rate of sCI and water vapour has been found to strongly depend on the structure of the Criegee Intermediate (Berndt et al., 2014c; Huang et al., 2015) and for the monoterpene-derived sCIs, the relative rate coefficients $k_{loss}$ / $k_{sCI+SO2}$ was found to be nearly independent of the relative humidity (Sipilä et al., 2014). The kinetic study of Huang et al. (2015) suggested that sCIs with more

complicated substitution groups (such as α-pinene derived sCIs) react with water slowly but react with SO$_2$ quickly, thus, supporting the reaction parameters used. Other possible loss paths of sCI are considered to be neglible. The studies of Vereecken et al. (2012 and 2014) show that a high substitution of CI and/or the other compound result in strong steric hindrance between the substituents, which effectively inhibits reactions between them. Thus, reactions between monoterpene-derived sCI and SVOCs are not favorable.

$$k_{loss} = k_{dec.} + k_{(sCI+H_2O)} \times \left[\mathrm{H_2O}\right] \qquad (4)$$

$$\frac{d[\mathrm{sCI}]}{dt} = Y_{sCI} \times k_{O_3+\alpha-pinene} \times \left[\mathrm{O_3}\right] \times \left[\alpha-\mathrm{pinene}\right] - k_{loss} \times \left[\mathrm{sCI}\right] - k_{sCI+SO_2} \times \left[\mathrm{SO_2}\right] \times \left[\mathrm{sCI}\right] \qquad (5)$$

The concentration of sCI was calculated according to equation 5, in which the values of the reaction rate coefficient ($k_{sCI+SO2}$), the sCI yield term ($Y_{sCI}$) and the loss term ($k_{loss}$) were taken from the TROPOS-LFT measurements (Sipilä et al., 2014). Those measurements were conducted at 50% relative humidity (RH) and the derived sCI yield

from the reaction between α-pinene and ozone was determined to be 0.15±0.07 and the ratio between the loss term and $k_{sCI+SO2}$ was $(2.0\pm0.4)\times10^{12}$ molecules cm$^{-3}$. Sipilä et al. (2014) also found that in the case of α-pinene and limonene the ratio $k_{loss}$ / $k_{sCI+SO2}$ was nearly independent of the relative humidity therefore we neglected the difference in RH of the experiments shown here (38%) compared to the experiments at TROPOS-LFT (50%). The temperature was 278 K in the CLOUD experiments whereas it was 293 K in the previous experiments. The

influence of temperature on the H$_2$SO$_4$ formation from the gas-phase reaction of monoterpene-derived sCIs has not yet been investigated. It is very likely that $k_{dec}$ is higher at higher temperatures, which would cause underestimation of the sulphuric acid concentration in the CLOUD simulations, where we are using the loss term derived from experiments performed at higher temperature. In the sCI yield experiments performed with acetone oxide the temperature influence on the ratio of $k_{dec}$ / $k_{sCI+SO2}$ was 2-fold when the temperature was increased by 10

K (Berndt et al., 2014c).





The minima and maxima of the sCI concentration were modelled with the upper and lower values for the yield and loss term so that the lower limit was calculated with a yield of 8% and a $k_{loss} / k_{sCI+SO2}$ ratio of $2.4 \times 10^{12}$ molecules cm$^{-3}$ while the upper limit was calculated with a yield of 22% and a $k_{loss} / k_{sCI+SO2}$ ratio of $1.6 \times 10^{12}$ molecules cm$^{-3}$ (Eq. (5)). For the calculations we needed to separate the terms $k_{loss}$ and $k_{sCI+SO2}$ from each other. As long as the ratio between the terms stays the same, the chosen values do not make a difference for the sulphuric acid concentration.

The concentration of sulphuric acid in the CLOUD chamber was modelled according to equation 6.

$$\frac{d[\mathrm{H_2SO_4}]}{dt} = k_{sCI+SO_2} \times [\mathrm{sCI}] \times [\mathrm{SO_2}] - (\mathrm{CS} + k_{wall\_loss} + k_{dil}) \times [\mathrm{H_2SO_4}] \qquad (6)$$

As an OH scavenger was used in the experiments, the only formation pathway for sulphuric acid was assumed to be the reaction between the sCI and $SO_2$. The production of sulphuric acid was calculated with the modelled sCI concentration, measured sulphur dioxide concentration and the reaction coefficient $k_{sCI+SO2}$ (Sipilä et al., 2014). The sulphur dioxide and ozone concentrations were kept constant during the experiments. Three loss processes were taken into account for sulphuric acid: the condensation sink (CS), the wall loss ($k_{wall\,loss}$) and the dilution ($k_{dil}$). The lifetime of sulphuric acid with respect to wall loss in the CLOUD chamber has been measured to be around 550s (Almeida et al., 2013; Duplissy et al., 2016; Rondo et al., 2014). The dilution rate due to injection of makeup gases into the chamber was $0.1 \times 10^{-3}$ s$^{-1}$.

The production rate of HOM in the CLOUD chamber (Eq. (7)) was calculated from the measured α-pinene and ozone concentrations, a reaction rate coefficient of $8.05 \times 10^{-17}$ cm$^3$ s$^{-1}$ (Atkinson et al., 2006) and an experimentally derived yield term (Ehn et al., 2014; Jokinen et al., 2015). The yield of HOM from the reaction between α-pinene and ozone was reported in recent studies. Ehn et al. (2014) obtained a yield of 7±3.5% in their experiments in the Jülich Plant Atmosphere Chamber while Jokinen et al. (2015) calculated a yield of 3.4% with an estimated uncertainty of −1.7/+3.4% from the experiments done in the TROPOS-LFT and Kirkby et al. (2016) reported a yield of 2.9% for the CLOUD experiments with and without ions. The same loss paths were taken into account in the modelled HOM concentration as for the sulphuric acid concentration. The lifetime of HOM was measured to be around 900s which is longer than the lifetime of sulphuric acid (Kirkby et al., 2016). All the values used in the modelling of sulphuric acid and HOM concentrations are shown in the Table 3.

$$\frac{d[HOM]}{dt} = Y_{HOM} \times k_{O_3+\alpha-pinene} \times [O_3] \times [\alpha-pinene] - (\mathrm{CS} + k_{wall\_loss} + k_{dil}) \times [HOM] \qquad (7)$$

30



**Table 3: Reaction rates, loss terms and yields used in simulations.**

| | |
|---|---|
| $k_{loss}$ / $k_{sCI+SO2}$ | $(1.6\text{-}2.4) \times 10^{12}$ molecules cm$^{-3}$ |
| $k_{O3+\alpha\text{-}pinene}$ | $8.05 \times 10^{-17}$ cm$^3$ s$^{-1}$ |
| $Y_{sCI}$ (**Sipilä** *et al.* **2014**) | 0.08-0.22 |
| $k_{wall\ loss\ (SA)}$ | $1.8 \times 10^{-3}$ cm$^3$ s$^{-1}$ |
| $k_{wall\ loss\ (HOM)}$ | $1.1 \times 10^{-3}$ cm$^3$ s$^{-1}$ |
| $k_{dil}$ | $0.1 \times 10^{-3}$ cm$^3$ s$^{-1}$ |
| $Y_{HOM}$ (**Ehn** *et al.* **2014**) | 0.035-0.105 |
| $Y_{HOM}$ (**Jokinen** *et al.* **2014**) | 0.017-0.068 |

## 3    Results

### 3.1 Reaction products from α-pinene ozonolysis

During the ozonolysis experiments of α-pinene, a simultaneous increase of the concentrations of sulphuric acid and HOM were observed. Several highly oxidised α-pinene oxidation products were observed between 220 and 620 Th (Fig. 1). All the HOM were detected as clusters with a nitrate ion ($NO_3^-$). As a result of the high cleanliness of the CLOUD chamber, the mass spectra consist mainly of the oxidation products and concentrations of contaminants were low. The most abundant HOM monomers, containing a $C_{10}$ carbon skeleton, had an O:C ratio between 0.7 and 1.1 whereas the most abundant HOM dimers, containing a $C_{20}$ carbon skeleton, had an O:C ratio around 0.6-0.8. The highest concentrations were observed from compounds identified as $C_{10}H_{14}O_7$, $C_{10}H_{15}O_8$, $C_{10}H_{14}O_9$, $C_{10}H_{15}O_{10}$, $C_{10}H_{16}O_{10}$ and $C_{19}H_{28}O_{11}$, which represent the majority of the total concentration of HOM. These compounds have also been found to be abundant in the boreal forest, when analysing the naturally charged ions (Ehn et al., 2012). The rest of the compounds taken into account in the concentration calculation are listed in Table A1.





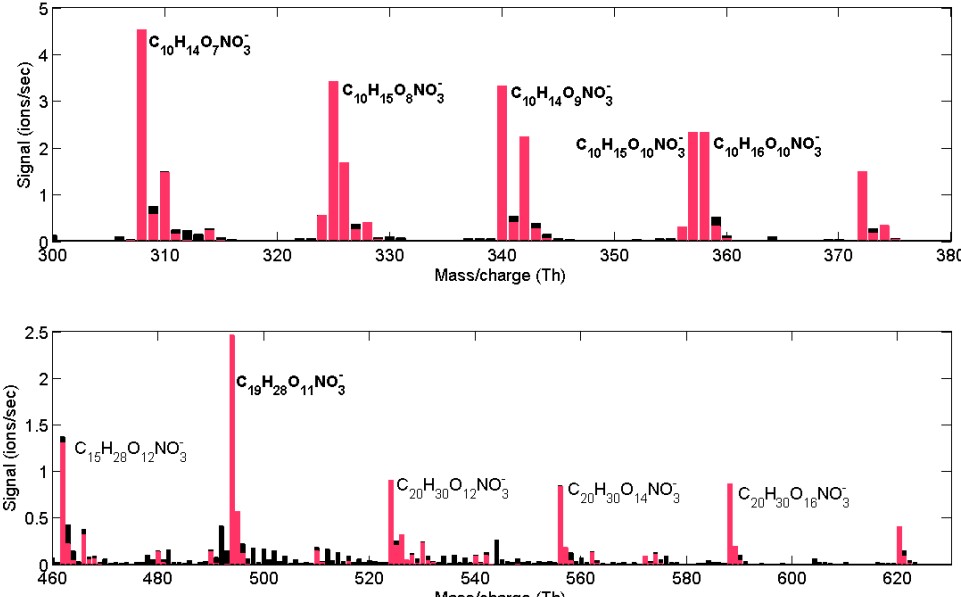

**Figure 1: HOM mass spectrum during an α-pinene ozonolysis experiment (C10 compounds in the upper panel and C20 compounds in the lower panel). The measured mass spectrum is depicted in black and the compounds identified as HOM are depicted in red. The elemental composition of the compounds with the highest concentrations are shown in the figure and the six most abundant compounds are labelled in bold face.**

During most of the experiments, clear particle formation and growth was observed shortly after the α-pinene injection was started. In Figure 2 the particle size distribution and precursor vapour concentrations during an example ozonolysis experiment are shown. In this experiment the α-pinene injection started at noon and the sulphuric acid and HOM concentrations started to increase immediately. The particle growth above 3 nm can be seen approximately 45 minutes after the injection. While the concentration of α-pinene continued to increase, the sulphuric acid and HOM concentrations reach their steady-state concentrations after one to two hours. The sulphuric acid reaches its steady-state concentration slightly before the HOM concentration reaches its maximum value which is expected due to its faster wall loss rate.





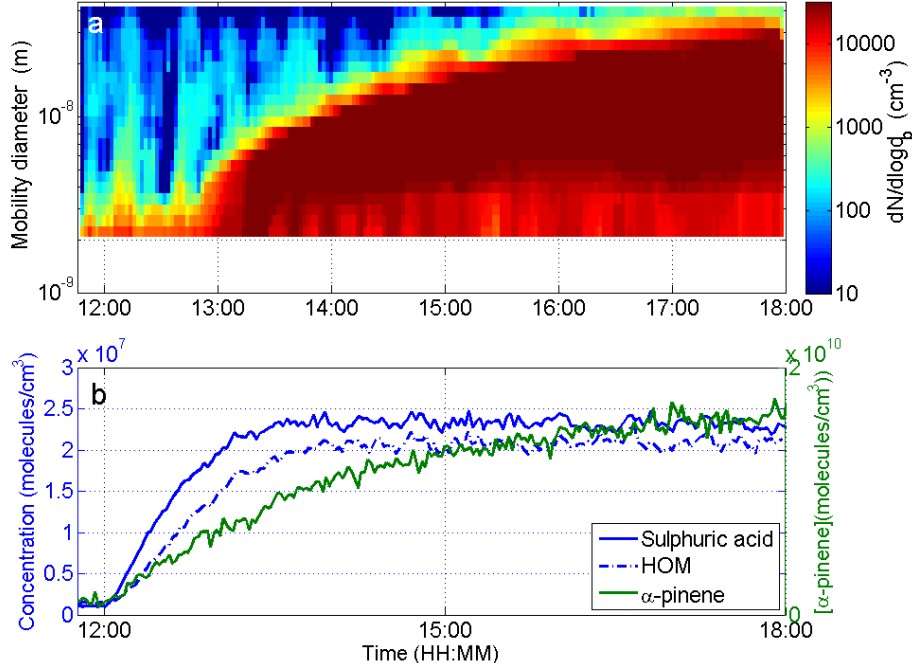

**Figure 2: Example of the size distribution (2-40 nm, measured by NAIS) of neutral particles (a) and the concentrations of sulphuric acid, HOM and α-pinene (b) during an ozonolysis experiment in the CLOUD chamber.**

The method presented in Sec. 2.3 was used to describe the temporal behaviour of the total HOM concentration. Since the total HOM is a sum of several molecules that are formed by the same autoxidation mechanism but possibly via various different intermediate steps, the time evolution of individual HOM molecules can differ from one to another. The time evolution of sulphuric acid and the most abundant HOM was studied in detail using mass

10 spectra integrated over 30s. The time evolution of the experiment with 600 pptv of α-pinene, 22 ppbv of $O_3$ and 67 ppbv of $SO_2$ is shown in Fig. 3. In our studies sulphuric acid concentration started to increase first followed by the concentration of $RO_2$ ($C_{10}H_{15}O_8$ and $C_{10}H_{15}O_{10}$). The closed shell monomers ($C_{10}H_{14}O_7$ and $C_{10}H_{14}O_9$) were formed right after $RO_2$. The most oxidised closed shell monomer of the selected HOM ($C_{10}H_{16}O_{10}$) and the dimer ($C_{19}H_{28}O_{11}$) took more than ten minutes to start increasing. The time evolution of the compounds might give us

15 information about the formation of the molecules. The rapid formation of the radicals and $C_{10}H_{15}O_8$ and $C_{10}H_{15}O_{10}$ implies that they are formed via autoxidation in which the peroxy radical undergoes oxidation by adding oxygen molecules stepwise. The selected dimer ($C_{19}H_{28}O_{11}$) formation starts clearly later, which supports the hypothesis that it forms from reaction of two $RO_2$ (Ehn et al., 2014; Jokinen et al., 2014). The carbon number 19 can be explained by loss of CO from $RO_2$ (Jagiella et al., 2000; Rissanen et al., 2014), followed by reaction with a 10-





carbon $RO_2$. The interesting feature in this data is that $C_{10}H_{16}O_{10}$ appears significantly later than most monomers, at the same time that the first dimer appears in the spectrum. This might indicate that this more highly oxidised product is also formed via bi-molecular reaction of two $RO_2$ radicals. The time evolution was similar in all the experiments. In the experiments with low α-pinene (80 pptv), the concentrations of $C_{19}H_{28}O_{11}$ and $C_{10}H_{16}O_{10}$ were

5    very low. It is also a possibility that the formation of dimers (and other compounds that appear later in the measurements) starts already earlier but the concentrations are just below the detection limit. Understanding the exact formation mechanisms of individual HOM compounds requires additional experiments and will be a topic of further studies.

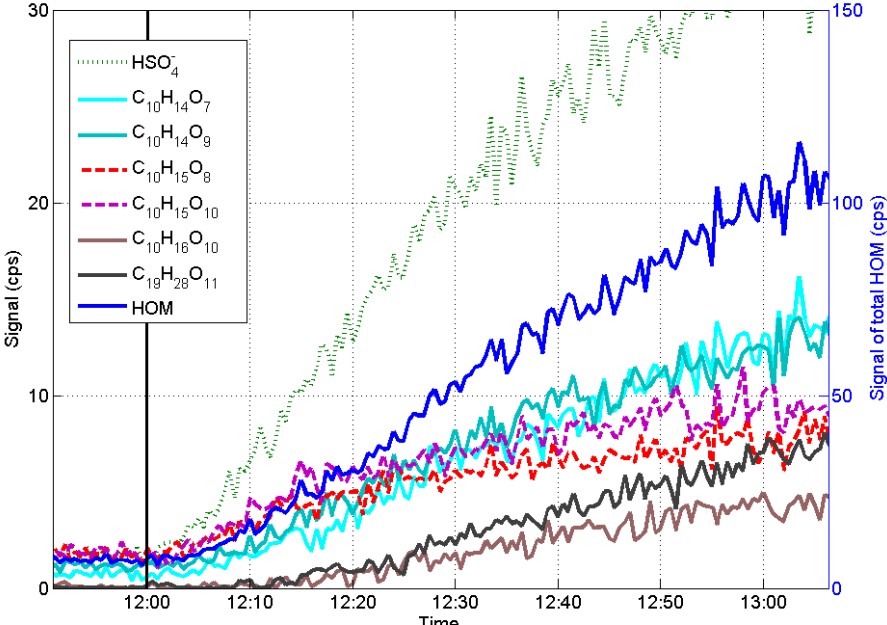

10    **Figure 3: The time evolution of bisulphate ion (green dotted line), total HOM (blue line) and the HOM signals of the highest concentrations with 30s time resolution in experiment with 600 ppt of α-pinene and 67 ppb of $SO_2$. $RO_2$ signals are shown with dashed lines. All HOM are detected as clusters with nitrate-ion. The black line shows the start of the experiment (i.e. α-pinene injection).**

15    In our experiments, we used $H_2$ to scavenge the OH. In the reaction of OH and $H_2$, water and H is produced and subsequently H can react with $O_2$ to form $HO_2$ (Eq. (8) and (9)). $HO_2$ can then react with $RO_2$ ending its autoxidation process (Ehn et al., 2014). This means that in the presence of $HO_2$ the HOM concentration can be lower because the organic compounds that react with $HO_2$ are not oxidised further into highly oxidised products. In these experiments, we did not have an instrument capable of measuring less oxidised products from α–pinene





ozonolysis. The relevance of these experiments to the atmosphere depends on the relative and absolute levels of all species participating in the autoxidation process, including $RO_2$, $HO_2$, and NO in both the experiment and the atmosphere. Jokinen et al. (2015) also used $H_2$ to scavenge OH while Ehn et al. (2014) used CO, both of which produce also H and then $HO_2$ (Eq. (10)). Thus, these experiments and the yield terms determined from them are

equally affected by $HO_2$. However, Jokinen et al. (2015) did also experiments with propane as OH scavenger, which does not produce $HO_2$, and found similar yields as with $H_2$. This implies that $HO_2$, produced by the scavenger reactions, does not significantly affect HOM formation.

$$OH + H_2 \rightarrow H_2O + H \qquad\qquad\qquad (8)$$

$$H + O_2 \rightarrow HO_2 \qquad\qquad\qquad (9)$$

$$OH + CO \rightarrow CO_2 + H \qquad\qquad\qquad (10)$$

### 3.2  The formation of sulphuric acid

The sulphuric acid formation in the CLOUD chamber was simulated as described in section 2.3. The measured steady-state concentrations varied between $4\times10^6$ and $2\times10^7$ molecules $cm^{-3}$ (Fig. 4). Sulphuric acid concentrations

were the highest in the experiments where also the α-pinene mixing ratio was the highest, around 600 pptv. The steady-state concentration was reached fast in the experiment at high $SO_2$ concentrations (~70 ppbv) whereas in the experiment with the same amount of α-pinene but significantly lower sulphur dioxide concentration (17 ppbv), the steady-state was reached an hour later (as expected). In the other three experiments the α-pinene mixing ratio was clearly lower (80 pptv and 170 pptv) and the increase of sulphuric acid concentration took more time and

continued throughout the whole experiment.

To compare the sCI oxidation with ambient sulphuric acid formation, we calculated the sulphuric acid produced at typical ambient OH concentration for otherwise similar conditions as in these experiments. The sulphur dioxide concentration was high in most of the experiments and with atmospherically relevant concentration of OH ($1\times10^6$ molecules $cm^{-3}$) the sulphuric acid concentration would be around $6.3\times10^8$ molecules $cm^{-3}$ ($SO_2$ 67 ppbv, reaction

rate constant $8.5\times10^{-13}$ $cm^3s^{-1}$ (Weber et al., 1996). With lower $SO_2$ concentration (17 ppbv) the OH-produced sulphuric acid would be around $1.6\times10^8$ molecules $cm^{-3}$. Thus, the sulphuric acid concentrations that resulted from sCI oxidation in these experiments were around 3% of what would be formed from OH oxidation in high $SO_2$ conditions and 10% in low $SO_2$ conditions at typical OH concentrations. It should be noted that in the atmosphere the mixture of gases is much more complex. In ambient conditions the α-pinene concentration is often less than

the concentration used in this calculation (600 pptv) but on the other hand in the atmosphere there are also other alkenes than α-pinene that can be oxidized to form sCI. Also the sulphur dioxide concentrations used in the experiments are relatively high for to most of the atmosphere, such as rural areas (Mikkonen et al., 2011).




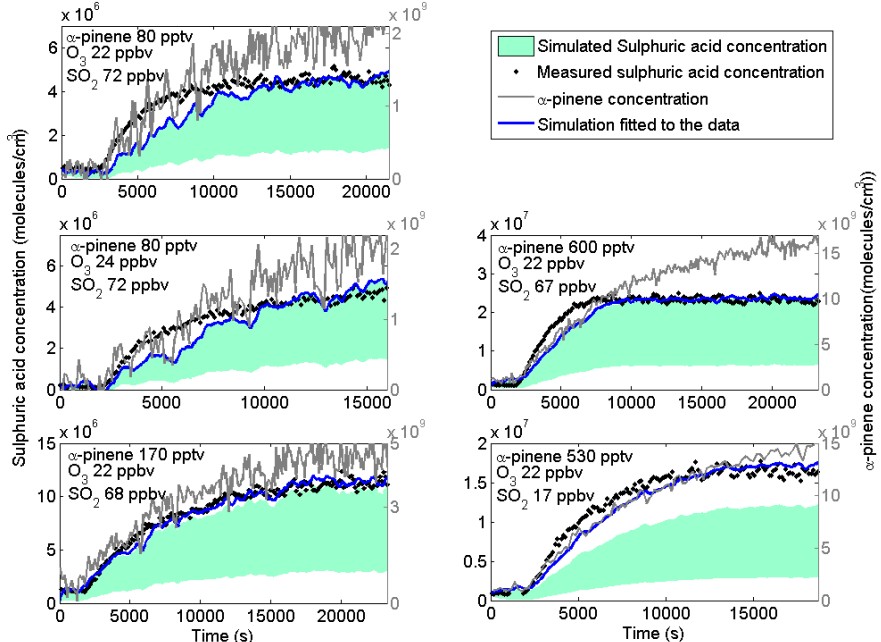

**Figure 4: Measured (black dots) and modelled (green shade shows the concentration with uncertainty) sulphuric acid concentrations formed from the oxidation of SO₂ by sCI in the CLOUD -chamber. The simulation fitted to the measured concentration is shown as a blue line and the α-pinene concentration is a grey line. Under conditions of atmospherical concentrations of OH (1×10⁶ molecules/cm³) the sulphuric acid concentrations would be significantly higher, around 6.3×10⁸ molecules/cm³ in the experiment with 67 ppb of SO₂ and 1.6×10⁸ molecules/cm³ in the experiment with 17 ppb of SO₂.**

In the simulations, the minimum and maximum concentrations were calculated from the upper and lower limits of given $k_{loss} / k_{sCI+SO2}$ and sCI yield term in Sipilä et al. (2014). In the CLOUD experiments, the measured sulphuric acid concentrations were at the upper range of the simulated concentrations in all the cases (Fig. 4). In the experiment where we had the least sulphur dioxide in the chamber, the measured concentration was slightly higher than the simulated one. The sulphuric acid was formed in the fast reaction between SO₂ and sCI, thus, the formation of sulphuric acid was strongly dependent on the formation of sCI. We calculated yield terms for the sCI in CLOUD experiments by fitting the model to the measured concentrations. When using a value of $1.6×10^{12}$ for the term $k_{loss} / k_{sCI+SO2}$ (lower end of the range given in Sipilä et al. 2014) the calculated yields of sCI were 22-24% for the experiments with higher concentration of SO₂ and 32% for the experiment with low SO₂. (Table 3). In the study done in the TROPOS-LFT the sCI yield from α-pinene oxidation was determined to be 8-22% (Sipilä et al., 2014). The simulated sulphuric acid concentration represent the measured concentrations well. When the α-pinene concentration was low, the measured α-pinene concentration was fluctuating which led to fluctuation in the simulated sulphuric acid concentration. In the experiment with a middle-range α-pinene concentration (170 pptv),





the upper bound of the simulated time evolution matched perfectly with the measured concentrations but in the other experiments the measured sulphuric acid concentration increased faster than the simulated concentration in the beginning but then stabilized at the upper level of the simulated concentrations. The difference is still small and mostly within the measurement uncertainty. In all the experiments, the simulated sulphuric acid concentration

followed the measured concentration very well after 10000 s (166 min) and thus the discrepancy cannot be explained by only one term. The simulation can be modified to match the measurements better if, for example, both the sCI yield term and condensation sink values are increased significantly. However, it seems unlikely that condensation sink for sulphuric acid would have such a large error. As mentioned earlier, the influence of temperature on the $H_2SO_4$ formation from the gas-phase reaction of monoterpene-derived sCIs has not been

investigated. It is likely that we underestimate the sulphuric acid concentration in the CLOUD experiments as we are using the loss term derived from experiments performed at higher temperature

### 3.3 The formation of HOM

The HOM formation in the CLOUD chamber was simulated as described in section 2.3. The measured steady-state concentrations of HOM varied between $2\times10^6$ and $2\times10^7$ molecules cm$^{-3}$ (Fig. 5). The respective α-pinene

and ozone concentrations were atmospherically relevant, so these HOM concentrations are similar to those found in ambient air (Sarnela et al., 2015; Yan et al., 2016). We calculated the HOM yield by fitting the model to the measured concentrations and obtained yields of 3.5-6.5% for the experiments in the CLOUD chamber (Table 3). In previous studies of α-pinene ozonolysis, yield terms for HOM formation have been experimentally determined. Ehn et al. (2014) measured a yield of 7±3.5% and Jokinen et al. (2015) a yield of 3.4% with an estimated

uncertainty of -50%/+100%. Kirkby et al. (2016) made a fit to both neutral and charged experiments and obtained a yield of 2.9% for the CLOUD experiments. The yields calculated in this study are in good agreement with all previous studies and the simulated time evolution  reproduce the measured concentrations very well.





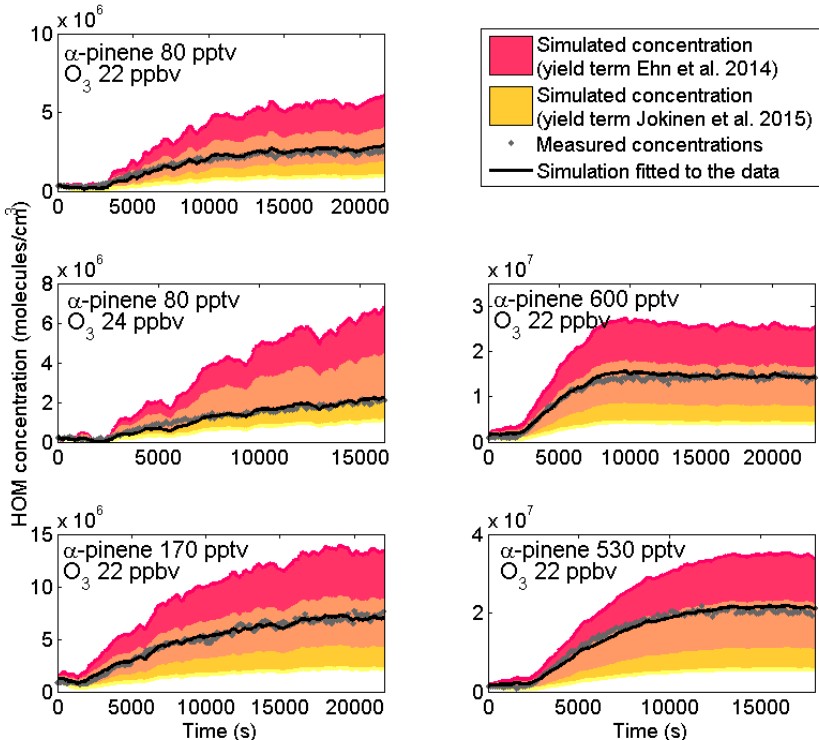

**Figure 5: Measured (grey dots) and simulated (shaded area show the concentration with uncertainty) HOM concentrations formed from ozonolysis of α-pinene in CLOUD-chamber. The simulation fitted to the measured concentration is shown as a black line. The modelled concentration with a yield term from Ehn et al. (2014) is shown in red shade while the concentration with the yield term from Jokinen et al. (2015) is shown in yellow shade. The overlapping area within the error estimates of these studies is coloured in orange shade.**

## 4    Conclusions

In this study we conducted several α-pinene ozonolysis experiments in an ultraclean environment, the CLOUD-chamber. These experiments were designed to be OH-radical free, thus allowing to study the formation of Highly Oxidized Molecules (HOM), from the ozonolysis of α-pinene. The other objective of this study was to observe the formation of sulphuric acid from the oxidation $SO_2$ by stabilized Criegee Intermediates. Both HOM and sulphuric acid concentrations were experimentally measured with a high resolution time-of-flight mass spectrometer by utilizing a highly selective chemical ionization method. To estimate the molar yield of the HOM and the sCI-yield





in our experiments, we used a 0-dimensional model with reaction parameters some of which were obtained from other recent publications on sCI and HOM formation (Ehn et al., 2014; Jokinen et al., 2015; Sipilä et al., 2014).

The formation of HOM was initiated immediately after an α-pinene injection into the chamber with a stable ozone concentration. We observed a consecutive formation of peroxy radicals, HOM monomer and dimer species, which
is in agreement with previous studies conducted in both, a laminar flow tube and in a continuously stirred flow reactor (Ehn et al., 2014; Jokinen et al., 2014). The simulated time evolution of the HOM followed the measured concentrations very precisely and the calculated yields from several experiments were in the range of 3.5-6.5%. The yields observed in the CLOUD chamber were within the range of previously published HOM yields for α-pinene ozonolysis (3.5-10.5% by Ehn et al., 2014, 1.7-6.8% by Jokinen et al., 2015, 1.2-5.8% Kirkby et al., 2016).

Sulphuric acid in the chamber was assumed to be solely produced via stabilized Criegee Intermediates reacting with the added $SO_2$. The formation of sulphuric acid started promptly after the α-pinene injection and the associated sCI formation. The measured concentration increased quickly, in some experiments even faster than was expected from the simulations. With a high $SO_2$ concentration (70 ppbv), the sCI yields were measured to reach 22-24%, which are on the upper edge of the values than found by Sipilä et al. (2014), i.e.15±7%. When the $SO_2$
concentration was considerably lower (17 ppbv) the sCI yield was higher (32%). These results are not denying that OH is the main daytime oxidizer of sulphur dioxide. In the presence of OH the role of sCI in the formation of sulphuric acid is relatively small but in dark conditions there can be considerable sulphuric acid formation due to sCI. The results of this study emphasize the potential importance of stabilized Criegee Intermediates in sulphuric acid formation, also in the presence of water vapour. In this paper we introduce a way to simulate the ozonolysis
products of α-pinene in a simple manner. The results indicate that the CLOUD experiments on α-pinene ozonolysis support the recently published chemistry of HOM and sCI formation, thus making the reaction parameters more reliable for broader modelling and theoretical use.

### Acknowledgements

We would like to thank CERN for supporting CLOUD with technical and financial resources. This research has
received funding from the EC Seventh Framework Programme (Marie Curie Initial Training Network 'CLOUD-ITN' no. 215072, MC-ITN 'CLOUD-TRAIN', no. 316662, and ERC-StG-ATMOGAIN (278277) and ERC-Advanced 'ATMNUCLE' grant no. 227463 and ERC-StG-GASPARCON (714621)), the German Federal Ministry of Education and Research (project nos 01LK0902A and 01LK1222A), the Swiss National Science Foundation (project nos 200020_135307, 200020_152907, 20FI20_149002 and 200021_140663), the Academy
of Finland Center of Excellence programme (grant no. 307331), the Academy of Finland (CoE project no. 1118615, LASTU project no. 135054), the Academy of Finland (296628), the Nessling Foundation, the Austrian Science Fund (FWF; project no. J3198-N21), the EU's Horizon 2020 research and innovation programme under the Marie Sklodowska-Curie (no. 656994), the Swedish Research Council, Vetenskapsrådet (grant no. 2011-5120), the Portuguese Foundation for Science and Technology (project no. CERN/FP/116387/2010), the Presidium of the
Russian Academy of Sciences and Russian Foundation for Basic Research (grants 08-02-91006-CERN and 12-02-91522-CERN), Dreyfus Award EP-11-117, the Davidow Foundation, the US National Science Foundation



(grants AGS1136479, AGS1447056, AGS1439551 and CHE1012293), US Department of Energy (grant DE-SC00014469) and the FP7 project BACCHUS (grant agreement 603445).

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



Appendix A

**Table A1: The elemental compositions and exact masses of most abundant isotopes of the HOM compounds that were added together to make the total HOM**

| Elemental composition | Exact mass (Th) |
|---|---|
| $C_7H_{10}O_4NO_3^-$ | 220.0463 |
| $C_5H_6O_6NO_3^-$ | 224.0048 |
| $C_5H_6O_7NO_3^-$ | 239.9997 |
| $C_8H_{12}O_7NO_3^-$ | 282.0461 |
| $C_8H_{12}O_8NO_3^-$ | 298.0416 |
| $C_{10}H_{14}O_7NO_3^-$ | 308.0623 |
| $C_9H_{12}O_8NO_3^-$ | 310.0416 |
| $C_{10}H_{16}O_7NO_3^-$ | 310.0780 |
| $C_8H_{12}O_9NO_3^-$ | 314.0365 |
| $C_{10}H_{14}O_8NO_3^-$ | 324.0572 |
| $C_{10}H_{15}O_8NO_3^-$ | 325.0651 |
| $C_9H_{12}O_9NO_3^-$ | 326.0365 |
| $C_{10}H_{16}O_8NO_3^-$ | 326.0729 |
| $C_9H_{14}O_9NO_3^-$ | 328.0521 |
| $C_{10}H_{14}O_9NO_3^-$ | 340.0521 |
| $C_9H_{12}O_{10}NO_3^-$ | 342.0314 |
| $C_{10}H_{16}O_9NO_3^-$ | 342.0678 |
| $C_{10}H_{14}O_{10}NO_3^-$ | 356.0471 |
| $C_{10}H_{15}O_{10}NO_3^-$ | 357.0549 |
| $C_9H_{12}O_{11}NO_3^-$ | 358.0263 |
| $C_{10}H_{16}O_{10}NO_3^-$ | 358.0627 |
| $C_{10}H_{14}O_{11}NO_3^-$ | 372.0420 |
| $C_9H_{12}O_{12}NO_3^-$ | 374.0212 |
| $C_{10}H_{16}O_{11}NO_3^-$ | 374.0576 |
| $C_{10}H_{14}O_{13}NO_3^-$ | 404.0318 |
| $C_{15}H_{28}O_{12}NO_3^-$ | 462.1464 |
| $C_{17}H_{24}O_{11}NO_3^-$ | 466.1202 |
| $C_{17}H_{26}O_{11}NO_3^-$ | 468.1359 |
| $C_{18}H_{26}O_{11}NO_3^-$ | 480.1359 |
| $C_{14}H_{20}O_{15}NO_3^-$ | 490.0686 |
| $C_{19}H_{28}O_{11}NO_3^-$ | 494.1515 |
| $C_{20}H_{32}O_{11}NO_3^-$ | 510.1828 |
| $C_{17}H_{26}O_{14}NO_3^-$ | 516.1206 |
| $C_{20}H_{30}O_{12}NO_3^-$ | 524.1621 |
| $C_{19}H_{28}O_{13}NO_3^-$ | 526.1414 |
| $C_{18}H_{26}O_{14}NO_3^-$ | 528.1206 |
| $C_{18}H_{28}O_{14}NO_3^-$ | 530.1363 |
| $C_{17}H_{26}O_{15}NO_3^-$ | 532.1155 |
| $C_{20}H_{30}O_{13}NO_3^-$ | 540.1570 |
| $C_{20}H_{32}O_{13}NO_3^-$ | 542.1727 |
| $C_{17}H_{26}O_{16}NO_3^-$ | 548.1105 |
| $C_{20}H_{30}O_{14}NO_3^-$ | 556.1519 |
| $C_{18}H_{28}O_{16}NO_3^-$ | 562.1261 |
| $C_{20}H_{30}O_{15}NO_3^-$ | 572.1468 |
| $C_{20}H_{32}O_{15}NO_3^-$ | 574.1625 |
| $C_{20}H_{30}O_{16}NO_3^-$ | 588.1418 |
| $C_{18}H_{28}O_{18}NO_3^-$ | 594.1159 |
| $C_{20}H_{30}O_{18}NO_3^-$ | 620.1316 |