# Peer review of "Measurement-model comparison of stabilized Criegee Intermediate and Highly Oxygenated Molecule production in the CLOUD chamber"

_Atmospheric Chemistry and Physics, 2017_

## Referee Comment (RC1) · Anonymous Referee #2 · 15 Oct 2017

Sarnela et al. presented a-pinene ozonolysis experimental results regarding modeled and observational comparisons of H2SO4 from sCI bi-radical reactions and HOM. This paper is clearly written and provides a comprehensive overview on a-pinene ozonolysis. The discussion about the current photochemical understanding of a-pinene ozonolysis contained in the 0D box model is well developed by comparing with observational results using a Api-Tof-CIMS instrument. The discussion outcomes will be highly beneficial to the research community so I support the publication of the manuscript. I would like to see some clarification on the argument in the conclusion suggesting the potential importance of the roles of the sCI sulfuric acid formation pathway during the night time when OH becomes absence. However, in this time, ozone should be also low

due to low photochemical activities. Therefore, it would be more informative to provide quantitative comparisons between the OH and the sCI pathways by calculating 24 hour H2SO4 productions from the both pathways using typical ozone and OH diurnal averages.

---

## Referee Comment (RC2) · Anonymous Referee #1 · 26 Oct 2017

The authors present measurements of highly oxidised molecules (HOMs) and sulfuric acid ($H_2SO_4$) using chemical ionization atmospheric pressure interface time-of-flight (CI-APi-TOF) mass spectrometry in $\alpha$-pinene ozonolysis experiments in the CLOUD chamber at CERN. The data are used in conjunction with model calculations to infer yields of HOMs and stabilized Criegee intermediates (sCI), which, in the presence of the OH radical scavenger $H_2$, are assumed to be responsible for the observed production of $H_2SO_4$.

The experiments, and analyses of the observations, are challenging, and the manuscript will be of interest to the atmospheric science community. However, I have

a number of comments listed below which ought to be addressed prior to publication, notably relating to the treatment of uncertainties and comparisons made between measurements and model simulations without any quantification.

Comments:

Page 1, line 9: Comma after 'Douglas'.

Page 1, line 34: Remove 'the' in '. . . through the ozonolysis. . .'.

Page 2, line 1: What do the authors mean by the term 'dynamic model'? A kinetic model, or simply calculation of expected production rates, might be a better description.

Page 2, line 1: Remove 'the' in '. . . for the HOM formation. . .'.

Page 2, line 2: Please be clear which measured concentrations the model is fit to – sCI concentrations have not been measured directly.

Page 2, line 4: Please quantify 'faster than simulated'.

Page 2, line 11: Remove 'a' in '. . . can act as a cloud . . .'.

Page 2, line 12: Replace 'have' with 'has' in 'A lot of effort have been. . .'.

Page 2, lines 28-30 and elsewhere: There is an inconsistency in the use of 'f' and 'ph' in the spellings for sulfur compounds. IUPAC recommend 'sulfur' over 'sulphur' and the authors should at least be consistent in their choice for all S compounds throughout the manuscript.

Page 2, line 33: The Mauldin III et al. reference does not explicitly demonstrate the reaction of sCI + SO2, but rather it is inferred as a possible explanation for their measurements.

Page 2, line 35: 'collision stabilized' to 'collisionally stablized'. Is the 2011 reference the most appropriate? There was an awareness of this behaviour prior to 2011.

Page 3, line 10: Please be clear which compound you are referring to in '. . .in which

the compound is . . .'. Perhaps something along the lines of '. . . in which the radical(s) produced after the initial oxidation are. . .'.

Page 3, line 10,13&elsewhere : Please be consistent is in use of 'oxidized' or 'oxidised'.

Page 3, line 13: Perhaps insert 'intramolecular' before 'H-shift'? It is not clear whether this mechanism was introduced by Ehn et al. as it has been known for many years in combustion chemistry.

Page 3, line 16: 'oxidixed'.

Page 3, line 20: Reactions of ROȞ2 with NO, RO2 and HO2 do not always form closed shell products.

Page 3, line 25: Remove 'the' in '. . . at least in the forested regions. . .'.

Page 4, Table 1: CLOUD description 'resindence'.

Page 4, line 21: The temperature stability seems extremely accurate for such a large volume. How is this achieved? Can the authors be sure there are no temperature gradients? A stability of 0.05 K is perhaps possible for the cooling system used for the chamber, but seems rather accurate for the entire volume of the chamber itself.

Page 5, line 8: Please consider changing 'cutting'.

Page 5, line 9: 'formed particles' to 'particles formed'.

Page 6, line 5: Is there an additional space before C6H8O7?

Page 6, line 13: What are the uncertainties in the calibration coefficients?

Page 7: Please provide some more details regarding the previous determinations of k(HOM), k(SA) and T(SA)/T(HOM). Given HOMs are a wide range of species, how representative is the value of k(HOM) determined experimentally? What are the ranges reported for k(HOM) and k(SA)? Are these upper and lower limits? What were the mean values and uncertainties? The statement that the rates are 'close to each other'

assumes that k(HOM) and k(SA) are each at the same point in their range, which is not necessarily the case. How would the results from this work be impacted if one were at its upper limit and the other at its lower limit? Please also provide further details on how the systematic uncertainties in H2SO4 and HOM concentrations were estimated, how these uncertainties compare to the simulations presented in Figures 4&5 and how the determinations of the yields are affected.

Page 8, lines 3-7: Was a yield term required to calculate the production of sCI? What are the references for the temperature-dependent rate coefficients? Are they also Atkinson et al. (2006)?

Page 8, line 7: 'is competed' to 'in competition with'.

Page 8, line 9: Is there any evidence for reaction with the water dimer?

Page 8, lines 16: Are the reaction parameters referred to those given in lines 25 & 26?

Page 8, lines 17-18: Which 'other compound' does this refer to? The sCI reaction partner? Is this relevant for discussion of reaction with SO2 or water?

Page 8, lines 20-22: Please consider some additional brackets in the equations.

Page 9, line 1: Please consider changing 'minima and maxima' to 'lower and upper limits' if this is what is being reported.

Page 10, line 9: Can the 'low' concentrations of contaminants be quantified?

Page 12, Figure 2: Are the data shown in (b) included in Figures 4&5? Is it necessary to reproduce the plots? Inclusion of the model simulations in Figures 4&5 make is more informative than the data shown in Figure 2.

Page 12, line 13: Can the statement 'formed right after RO2' be quantified? How soon is 'right after'?

Page 14, line 16: Quantify 'fast'.

Page 14, line 18: What was the expectation based on? If this uses model simulations can these be shown?

Page 14, line 24: Is there a closing parenthesis missing?

Page 14, line 32: Remove 'to' following '...relatively high for ...'.

Page 15, Figure 4: Which parameters were varied in the fitted simulation?

Page 15, line 3: Is the hyphen needed in '-chamber'?

Page 15, line 11: Quantify 'slightly higher'.

Page 16, lines 1&7&22: Quantify the terms 'matched perfectly', 'increased significantly' and 'reproduce the measured concentrations very well'.

Page 16, line 11: Is there a full stop missing at the end of the sentence?

Page 17, Figure 5: What does the colour in the plot represent that isn't listed in the legend? Is it the overlap between the simulations using Ehn results with those of Jokinen? Uncertainty in the fit? Which parameters were fitted?

Page 18, line 21: Which reaction parameters specifically? What is meant by 'broader modelling'?

References: Several formatting issues (e.g. page 19 line 19, page 20, line 11) and with subscripts.

---

## Author Comment (AC1) · 19 Dec 2017

**Responses to referees "Measurement-model comparison of stabilized Criegee Intermediate and Highly Oxygenated Molecule production in the CLOUD chamber" by Sarnela et al.**

Referee's comments in black
Author's answers in blue
The revised manuscript with track changes can be found below the responses to the referees.

**Responses to Referee #1**

The authors present measurements of highly oxidised molecules (HOMs) and sulfuric acid ($H_2SO_4$) using chemical ionization atmospheric pressure interface time-of-flight (CI-APi-TOF) mass spectrometry in α-pinene ozonolysis experiments in the CLOUD chamber at CERN. The data are used in conjunction with model calculations to infer yields of HOMs and stabilized Criegee intermediates (sCI), which, in the presence of the OH radical scavenger H2, are assumed to be responsible for the observed production of $H_2SO_4$.

The experiments, and analyses of the observations, are challenging, and the manuscript will be of interest to the atmospheric science community. However, I have a number of comments listed below which ought to be addressed prior to publication, notably relating to the treatment of uncertainties and comparisons made between measurements and model simulations without any quantification.

We thank the referee for very precise comments. These comments improved the manuscript considerably.

Comments:

Page 1, line 9: Comma after 'Douglas'.

Page 1, line 34: Remove 'the' in ': : : through the ozonolysis: : :'.

We corrected these mistakes.

Page 2, line 1: What do the authors mean by the term 'dynamic model'? A kinetic model, or simply calculation of expected production rates, might be a better description.

We changed the description to kinetic model.

Page 2, line 1: Remove 'the' in ': : : for the HOM formation: : :'.

Corrected as suggested.

Page 2, line 2: Please be clear which measured concentrations the model is fit to – sCI concentrations have not been measured directly.

We added "measured sulfuric acid concentration" to clarify the sentence.

Page 2, line 4: Please quantify 'faster than simulated'.

We added a sentence to quantify the results: "In those experiments the simulated and measured concentrations met when the concentration reached plateau but plateau was reached 20-50 minutes later in simulations."

Page 2, line 11: Remove 'a' in ': : : can act as a cloud : : :'.

Page 2, line 12: Replace 'have' with 'has' in 'A lot of effort have been: : :'.

We corrected these errors.

Page 2, lines 28-30 and elsewhere: There is an inconsistency in the use of 'f' and 'ph' in the spellings for sulfur compounds. IUPAC recommend 'sulfur' over 'sulphur' and the authors should at least be consistent in their choice for all S compounds throughout the manuscript.

We changed sulphur to sulfur according to IUPAC recommendation.

Page 2, line 33: The Mauldin III et al. reference does not explicitly demonstrate the reaction of sCI + SO2, but rather it is inferred as a possible explanation for their measurements.

This is true. We tried to bring this up by stating "the potential of the atmospheric relevance of sCI+SO2". We also added a reference of Berndt et al. 2012.

Page 2, line 35: 'collision stabilized' to 'collisionally stablized'. Is the 2011 reference the most appropriate? There was an awareness of this behaviour prior to 2011.

In revised manuscript we refer also to Herron et al. 1982, which is indeed important to mention.

Page 3, line 10: Please be clear which compound you are referring to in ': : :in which the compound is : : :'. Perhaps something along the lines of ': : : in which the radical(s) produced after the initial oxidation are: : :'.

Page 3, line 10,13&elsewhere : Please be consistent is in use of 'oxidized' or 'oxidised'.

Page 3, line 13: Perhaps insert 'intramolecular' before 'H-shift'? It is not clear whether this mechanism was introduced by Ehn et al. as it has been known for many years in combustion chemistry.

Page 3, line 16: 'oxidixed'.

We corrected these errors as guided.

Page 3, line 20: Reactions of RO$\check{A}$n2 with NO, RO2 and HO2 do not always form closed shell products.

We changed the verb to "can" instead of "will" to remove the meaning that closed shell products are the only result.

Page 3, line 25: Remove 'the' in ': : : at least in the forested regions: : :'.

Page 4, Table 1: CLOUD description 'resindence'.

Corrected.

Page 4, line 21: The temperature stability seems extremely accurate for such a large volume. How is this achieved? Can the authors be sure there are no temperature gradients? A stability of 0.05 K is perhaps possible for the cooling system used for the chamber, but seems rather accurate for the entire volume of the chamber itself.

The chamber is designed to achieve a high standard of temperature stability. The chamber is surrounded by an insulated thermal housing. The CLOUD chamber temperature is controlled by precisely regulating the temperature of air circulating in the space between the chamber and the thermal housing and two fans run in counter flow to achieve efficient turbulent mixing of the gases and the ions in the chamber. Forty temperature sensors monitor the temperature of the chamber's external wall and a string of 5 PT100 temperature sensors is placed at midplane level inside the chamber, at distances of 0.05, 0.2, 0.4, 0.8, and 1.2 m from the chamber wall. The design of the chamber is well described in Duplissy et al. 2016 and we added this reference to temperature stability.

Page 5, line 8: Please consider changing 'cutting'.

Page 5, line 9: 'formed particles' to 'particles formed'.

Page 6, line 5: Is there an additional space before C6H8O7?

Corrected.

Page 6, line 13: What are the uncertainties in the calibration coefficients?

The uncertainty for sulfuric acid concentration is estimated to be +50%/−33%. This estimate is based on a comparison of sulfuric acid measurements with a CIMS and a calibrated $H_2SO_4$ generator (Kirkby et al. 2016). Sulphuric acid calibration is taken into account in the systematic scale uncertainties of both sulfuric acid and HOM concentrations As we do not have direct calibration for HOM, we used the same calibration coefficient with additional terms as explained in section "2.3 Estimation of HOM sensitivity".

Page 7: Please provide some more details regarding the previous determinations of k(HOM), k(SA) and T(SA)/T(HOM). Given HOMs are a wide range of species, how representative is the value of k(HOM) determined experimentally? What are the ranges reported for k(HOM) and k(SA)? Are these upper and lower limits? What were the mean values and uncertainties? The statement that the rates are 'close to each other' assumes that k(HOM) and k(SA) are each at the same point in their range, which is not necessarily the case. How would the results from this work be impacted if one were at its upper limit and the other at its lower limit? Please also provide further details on how the systematic uncertainties in H2SO4 and HOM concentrations were estimated, how these uncertainties compare to the simulations presented in Figures 4&5 and how the determinations of the yields are affected.

The k(HOM) values presented earlier are not experimental but computational. The general framework for computing collision frequencies using quantum chemical data is discussed in detail e.g. by Garden et al. (2009). The collision frequency depends on the dipole moment and polarisability, both which can be obtained (as discussed in the reference above) fairly accurately with quantum chemical methods (much more accurately than e.g. reaction rate or equilibrium coefficients). For precisely known chemical structures, collision rates computed with high-level quantum chemistry are (according to the above reference) accurate to about 5% (Garden et al. 2009), while more modest levels of theory lead to accuracies of about 20%. For the HOM, the problem is that the structures are not precisely known. Fortunately, the larger the molecule, the less dependent the collision rate is on the precise chemical structure (e.g. location of functional groups with respect to each other). This is because the collision rate depends both on the dipole moment, which varies significantly between structural (and conformational) isomers, and the polarisability, which depends much less on the particular molecular structure (and more on the molecular size, general type of functional groups present, etc). The larger the molecule, the more important the relative contribution of the polarisability compared to the dipole moment. The values in the Ehn et al. (2014) study were based on three representative HOM structures (in line with the general mechanism presented in the paper), with dipole moments varying from 2.1 to 6.3 Debye, and (isotropic) polarizabilities varying from 20.7 to 21.37 Bohr$^3$. The relatively large variation dipole moments and small variation in polarizabilities illustrates the issue described above. Ehn et al. performed calculations at a quite modest level of theory, corresponding to the "inherent" error margin of 20% discussed by Garden et al. (2009). Assuming as an upper limit that the dipole moments of HOM might vary from 1 to 9 Debye (the upper limit corresponding to dipole moments computed for acid-base clusters with proton transfer - likely much larger than the dipole moment of single any oxidised organic molecule), and that the polarizabilities of HOM monomers with ca 10 C atoms and 7-10 O atoms (i.e. masses roughly around 250 amu) varies between 15 and 25 Bohr$^3$ (this relatively wide range includes the 20% error from the computational method), they obtained collision frequencies with $HNO_3*NO_3^-$ varying between $1.1\times10^{-9}$ and $2.6\times10^{-9}$ cm$^3$s$^{-1}$ mol$^{-1}$, and collision frequencies with $NO_3^-$ varying between $1.5\times10^{-9}$ and $3.4\times10^{-9}$ cm$^3$s$^{-1}$ mol$^{-1}$. Varying the assumed HOM mass between 150 and 350 amu further extended the upper limit of the range to $3.7\times10^{-9}$ while the lower limit remained the same with two-digit precision at $1.1\times10^{-9}$.

This given range ($1.1 \times 10^{-9}$ to $3.7 \times 10^{-9}$) thus represents the feasible maximum and minimum collision rates with $NO_3^-(HNO_3)_{0-1}$ for HOM monomers - much higher or much lower values would require very exotic chemical structures (even more so than the polyhydroperoxides already postulated). Mean values cannot be meaningfully computed without more detailed structural information (including actual yields of different structural isomers corresponding to the same elemental composition). However, the above calculations indicate that k(HOM) is unikely to differ from k(SA) in either direction by much more than a factor of 2 for a quite wide range of potential HOM structures and this is referred as "close to each other" in the text.

We presented HOM yields of 3.5-6.5% with an estimated uncertainty of -60%/+100%. If k(HOM) and k(SA) were at the opposite sides of their range, k(SA)/k(HOM) would be 0.53 or 1.67 instead of our approximation of 1. This would change our yields to values 1.9-10.8%. These yields are still within our uncertainty and close to the yields that Ehn et al. (3.5-10.5%) and Jokinen et al. (1.7-6.8%) have previously presented.

The systematic scale uncertainty for $[H_2SO_4]$ is estimated to be +50%/−33%. This estimate is based on a comparison of $[H_2SO_4]$ measurements with a CIMS and a calibrated $H_2SO_4$ generator. After consideration we increased the HOM yield estimated uncertainty from +100%/-50% to +100%/-60%. The systematic uncertainties for [HOM] have the following sources and fractional errors ($1\sigma$): sulfuric acid calibration (50%), charging efficiency of HOMs in the ion source (25%), mass dependent transmission efficiency (50%) and sampling line losses (20%). This results in an overall systematic scale uncertainty for [HOM] of +80%/−45%. The uncertainty in the HOM yield from ozonolysis is estimated by adding the [HOM] uncertainty in quadrature with the errors for α-pinene (10%), $O_3$ (10%), HOM wall loss rate (6%) and rate constants (35% for the α-pinene $O_3$ reaction). This results in a mean estimated uncertainty in HOM yield of +100%/−60%. This explanation can be found in Kirkby *et al.* 2016 which describes the same CLOUD experiments and that we have referred I the text. We described the uncertainty sources in the text: "We estimated an uncertainty of +80%/-45% for HOM concentrations taking the sulfuric acid calibration, charging efficiency, mass dependent transmission efficiency calibration and sampling line losses into consideration (Jokinen et al., 2015; Kirkby et al., 2016). The uncertainty for HOM yield arises from the uncertainties of α-pinene concentration, $O_3$ concentration, HOM wall loss rate and rate constants. This results in a mean estimated uncertainty in HOM yield of +100%/−60%."

Page 8, lines 3-7: Was a yield term required to calculate the production of sCI? What are the references for the temperature-dependent rate coefficients? Are they also Atkinson et al. (2006)?
There is a yield term in sCI production as shown in Eq. 5. In Atkinson et al. (2006) there is an equation for temperature-dependent rate coefficient ($8.05 \times 10^{-16}$ exp(-640/$T$)). The equation can be found in updated data sheet, and that is now clearly added to the text:
"(Atkinson et al., 2006, updated data sheet can be found: http://iupac.pole-ether.fr/htdocs/datasheets/pdf/ Ox_VOC8_O3_apinene.pdf))."

Page 8, line 7: 'is competed' to 'in competition with'.
Corrected.

Page 8, line 9: Is there any evidence for reaction with the water dimer?
$CH_2OO$ has been seen to react fast with water dimer (Berndt *et al.* 2014) but similar results have not been measured with alpha-pinene.

Page 8, lines 16: Are the reaction parameters referred to those given in lines 25 & 26?

This was imprecisely written, we corrected:"supporting the reaction parameters achieved by Sipilä et al. (2014)".

Page 8, lines 17-18: Which 'other compound' does this refer to? The sCI reaction partner? Is this relevant for discussion of reaction with SO2 or water?

In this we mean the reaction partner and we changed the term as "reaction partner". With these sentences we wanted to explain that sCI does not react with SVOCs that are in the chamber as written in the subsequent sentence which leaves water as the only competitive reagent with $SO_2$.

Page 8, lines 20-22: Please consider some additional brackets in the equations.

We added brackets for clarification.

Page 9, line 1: Please consider changing 'minima and maxima' to 'lower and upper limits' if this is what is being reported.

Changed as suggested.

Page 10, line 9: Can the 'low' concentrations of contaminants be quantified?

With this sentence we meant that as it can be seen in Fig. 1, most of the compounds that we see in the spectra are HOM (or sulfuric acid in lower mass range). Surely there are other identified and unidentified compounds in the spectra but major peaks are SA or HOM. We cannot quantify the rest of the compounds without calibration.

Page 12, Figure 2: Are the data shown in (b) included in Figures 4&5? Is it necessary to reproduce the plots? Inclusion of the model simulations in Figures 4&5 make is more informative than the data shown in Figure 2.

Yes the data is also shown in Fig 4. and 5. With this plot we wanted to show how the particle formation is linked to the vapours. The particle concentrations are not shown in any other figures. The comparison of source vapours and the particle concentrations is in our opinion easier when they are plotted in the same figure.

Page 12, line 13: Can the statement 'formed right after RO2' be quantified? How soon is 'right after'?

We changed the sentence to" The formation of closed shell monomers ($C_{10}H_{14}O_7$ and $C_{10}H_{14}O_9$) started a few minutes after the $RO_2$." to give the order of magnitude of the time.

Page 14, line 16: Quantify 'fast'.

This was unclearly written. We changed it to "in two hours".

Page 14, line 18: What was the expectation based on? If this uses model simulations can these be shown?

We took the sentence as expected off since no additional simulations were used.

Page 14, line 24: Is there a closing parenthesis missing?

Page 14, line 32: Remove 'to' following ': : :relatively high for : : :'.

Corrected as suggested.

Page 15, Figure 4: Which parameters were varied in the fitted simulation?

We specified "simulation with fitted yield term to the measured concentration" to make this clear.

Page 15, line 3: Is the hyphen needed in '-chamber'?

Hyphen is removed.

Page 15, line 11: Quantify 'slightly higher'.

We added the exact concentrations to the text "(measured $1.6\times10^7$ molecules cm$^{-3}$, upper range simulated concentration $1.2\times10^7$ molecules cm$^{-3}$)".

Page 16, lines 1&7&22: Quantify the terms 'matched perfectly', 'increased significantly' and 'reproduce the measured concentrations very well'.

Thank you for these remarks, clearly we have described some of our results vaguely. We added following sentences to make our statements more precise:

"the simulated time evolution matched perfectly with the measured concentrations, so that the trend in measured and simulated concentration was identical and the difference of simulated concentration from measured concentration did not exceed 30%."

"increased significantly (two-fold increase in both condensation sink and sCI yield)."

"The highest difference between simulated and measured concentration was 40% but in most experiments the simulated and measured concentrations matched within 20% difference."

Page 16, line 11: Is there a full stop missing at the end of the sentence?

Full stop added.

Page 17, Figure 5: What does the colour in the plot represent that isn't listed in the legend? Is it the overlap between the simulations using Ehn results with those of Jokinen? Uncertainty in the fit? Which parameters were fitted?

It is the overlap and it is explained in the caption. "The overlapping area within the error estimates of these studies is coloured in orange shade." We also added "The simulation with fitted yield term…" to the caption.

Page 18, line 21: Which reaction parameters specifically? What is meant by 'broader modelling'?

We specified the parameters "thus making the experimentally determined yield and loss terms more reliable for following modelling and theoretical use"

References: Several formatting issues (e.g. page 19 line 19, page 20, line 11) and with subscripts.

Errors corrected.

**Responses to Referee #2**

Sarnela et al. presented a-pinene ozonolysis experimental results regarding modelled and observational comparisons of H2SO4 from sCI bi-radical reactions and HOM. This paper is clearly written and provides a comprehensive overview on a-pinene ozonolysis. The discussion about the current photochemical understanding of a-pinene ozonolysis contained in the 0D box model is well developed by comparing with observational results using a Api-Tof-CIMS instrument. The discussion outcomes will be highly beneficial to the research community so I support the publication of the manuscript. I would like to see some clarification on the argument in the conclusion suggesting the potential importance of the roles of the sCI sulfuric acid formation pathway during the night time when OH becomes absence. However, in this time, ozone should be also low due to low photochemical activities. Therefore, it would be more informative to provide quantitative comparisons between the OH and the sCI pathways by calculating 24 hour H2SO4 productions from the both pathways using typical ozone and OH diurnal averages.

We thank the referee for the positive appraisal of the manuscript. We also thank for the fine suggestion how to better show the importance of sCI in SA formation during night time. We added a new paragraph and a figure to the manuscript with the 24 hour $H_2SO_4$ production calculation.

"To get more insight in the sulfuric acid production with ambient concentrations we calculated 24 hour production of sulphuric acid from OH and sCI oxidation pathways. We used typical spring – summer time concentrations of precursors in boreal forest: measured OH concentrations (medians of event day concentrations from late March to early June, Petäjä et al., 2008), measured $O_3$, $SO_2$, (medians of concentrations from April to June in 2013, Smart-SMEAR https://avaa.tdata.fi/web/smart/smear, Junninen et al., 2009), measured monoterpene concentrations (concentrations measured in July 2004, Rinne et al., 2005) and calculated condensation sink values (median of data from April to June 2013 Junninen et al., 2009; Kulmala et al., 2001). The 24 hour sulfuric acid productions were calculated with sCI yield of 22% and the results are shown in Fig. 5a. During the daytime the sulfuric acid produced by OH dominates but during night time both of the production pathways are important. In this example with conditions of boreal forest the $SO_2$ concentration is significantly lower than in our experiments (Fig 5d, around 0.1 ppb). Ozone concentrations are the lowest during early morning being around 35 ppbv while the concentration reach 43 ppbv in the evening. The importance of sCI in the sulfuric acid production strongly depends on the monoterpene concentrations: in this example the monoterpene concentration is the highest during early hours and at that time the sulfuric acid concentration reaches $4\times10^4$ molecules/cm$^3$. We calculated sCI yield term of 32% in our experiment with low $SO_2$ and if we use that yield term in the calculation the highest sulfuric acid concentration is $6\times10^4$ molecules/cm$^3$."

[Figure]

**Figure 5. Example of sulfuric acid concentration produced by OH and sCI in ambient boreal forest conditions (a). The precursor gas concentrations and condensation sink used are shown in plots b-e.**

[revised manuscript text omitted]

---

## Author Response (AR2)

Dear Co-editor,

We thank for your technical comments. In this revised manuscript we have taken them into consideration, updated the affiliations and reference list and checked our spelling. The changes are highlighted.

A point-by-point reply to the comments:

On page 16 line 5 you refer to "dynamic model" but I believe you had changed this to "kinetic model" elsewhere. This is corrected to "kinetic model".

On page 16 line 30-32. I wonder whether you can state more directly that this implies that k_dec dominate k_loss. It may be confusing to readers to have a term that contains the concentration of water but than state that water concentration is irrelevant.

This is a very good point. We added "under these conditions thermal decomposition dominates the loss mechanism of sCI and the reaction with water vapour is less important."

On page 25 line 28. I believe you are missing "the" between "that" and "condensation sink". The is added.

On p.26 line 29-30 (and page 28 line 27) . Could you provide a mechanistic suggestion why the SCI yield would depend on the SCI concentration or does this reflect experimental uncertainties?

[revised manuscript text omitted]